# Spiking Brain Compression: Post-Training Second-order Compression for Spiking Neural Networks

## Abstract

Spiking Neural Networks (SNNs) have emerged as a new generation of energy-efficient neural networks suitable for implementation on neuromorphic hardware. As neuromorphic hardware has limited memory and computing resources, weight pruning and quantization have recently been explored to improve SNNs' efficiency. State-of-the-art SNN pruning/quantization methods employ multiple compression and training iterations, increasing the cost for pre-trained or very large SNNs. In this paper, we propose a novel one-shot post-training compression framework, Spiking Brain Compression (SBC), that extends the classical Optimal Brain Surgeon (OBS) method to SNNs. SBC replaces the current-based objective found in common layer-wise compression method with a spike train-based objective whose Hessian is cheaply computable, allowing a single backward pass to compress synapses and analytically rescale the rest. Applying SBC to SNN pruning and quantization, Our experiments on models trained with neuromorphic datasets (N-MNIST, CIFAR10-DVS, DVS128-Gesture) and large static datasets (CIFAR-100, ImageNet) show state-of-the-art results for SNNs one-shot post-training compression methods, with single-digit to double-digit accuracy gains compared to ANN methods applied to SNNs. Combined with finetuning, SBC is also competitive with the accuracy of costly iterative methods, while cutting compression time by two orders of magnitude.

## 1 Introduction

Spiking Neural Networks (SNNs) have garnered significant attention as a promising power-efficient alternative to traditional Artificial Neural Networks (ANNs). They are comprised of spiking neurons that communicate through discrete spikes, analogously to their biological counterparts. This sparse, spike-based neuron dynamic enjoys benefits such as low computational power and compatibility with temporal data (Yamazaki et al., 2022), as well as high power efficiency when implemented on neuromorphic chips like True North (Akopyan et al., 2015), Loihi 1 and 2 (Orchard et al., 2021), or SpiNNaker (Furber et al., 2014). However, as these chips have limited computing power and memory, finding ways to improve SNNs' efficiency on neuromorphic hardware is an important research question (Neil et al., 2016). In this work, we focus on model compression for SNNs, specifically post-training pruning and quantization.

**SNN pruning techniques** Several methods for pruning SNNs have emerged in the past five years. (Shi et al., 2024) finds that aggressive unstructured pruning ($\geq 90\%$ sparsity) can yield 10× efficiency increase. (Guo et al., 2020) proposed an unsupervised online pruning technique for SNNs. (Kim et al., 2022) explored the lottery ticket hypothesis (LTH) for SNN and used it as the basis for their unstructured weight pruning method. However, most state-of-the-art (SOTA) SNN pruning methods either require the pruning step to happen during training or adopt an iterative pruning technique, where multiple pruning and training iterations are performed. Such approaches are computationally expensive, especially for pre-trained, deep spiking neural networks like SEW-ResNet 152 (Fang et al., 2022) and Spiking Transformers (Yao et al., 2023; Zhou et al., 2022). With the adaptation of the transformer architecture for SNNs, the scale of future state-of-the-art SNN models

could cause the retraining cost to become prohibitively large. Therefore, a more computationally efficient pruning algorithm is urgently needed for the deployment of future next-generation SNNs.

**SNN quantization techniques** Efforts to apply quantization to SNNs on neuromorphic hardware have recently gained traction. There are two main categories in neural network quantization: Quantization-Aware Training (QAT) and Post Training Quantization (PTQ). QAT allows models to adapt to the quantization through retraining, and there have been several articles written on QAT for SNNs, such as (Lui & Neftci, 2021; Furber et al., 2022; Wei et al., 2025). However, PTQ for SNNs is still relatively underdeveloped, especially compared to its ANN counterparts, where the desire to reduce the size of Large Language Models (LLMs) to run on smaller or even edge computing hardware has inspired many algorithms, such as AdaQuant (Jhunjhunwala et al., 2021), Optimal Brain Quantizer (OBQ) (Frantar et al., 2023b), and GPTQ (Frantar et al., 2023a).

**Other efficiency improvement techniques** Some other notable methods of improving SNN efficiency include knowledge distillation (Xu et al., 2023; Kushawaha et al., 2020), where a larger model is used to train a smaller model, spike rate reduction (Deng et al., 2020; Na et al., 2022; Han & Roy, 2020) and hardware-specific optimizations (Padovano et al., 2024; Fang et al., 2020). Due to space limitations, we do not cover these methods in depth here; they are orthogonal and can be used to complement pruning/quantization, as seen in recent works (Shymyrbay et al., 2023; Chowdhury et al., 2021).

We discovered that the naive application of ANN one-shot post training compression methods like Optimal Brain Compression (Frantar et al., 2023b) to SNNs yields suboptimal results, due to the disconnect between ANN methods' objective function and SNNs' layer-wise output spike train. To address this research gap, we derived an innovative objective function with the Van Rossum Distance (VRD) (Rossum, 2001) of output spike trains, referred to as surrogate membrane potential (SMP). By statistically estimating SMP's Hessian from a calibration dataset, SBC maintains OBC's computational cost while providing tighter compression guarantees.

Based on SMP, we introduce Spiking Brain Compressionß, a layer-wise compression algorithm using SMP to compress pre-trained SNNs in one shot. We validate SBC on pruning and quantization tasks with neuromorphic benchmarks and large-scale static datasets, demonstrating efficient, accurate compression of SNNs.

The main contributions of this work can be summarized as follows:

- We formalize a VRD–based loss for layer-wise SNN compression and derive an efficient Hessian that can be sampled from a calibration dataset.
- We develop *Spiking Brain Compression (SBC)*: a one-shot second-order SNN compression algorithm that works with complex SNNs like Spiking ResNets and Spiking Transformers.
- We validate SBC empirically across domains, demonstrating aggressive compression (97%) with minimal accuracy loss ($\leq 1\%$) on neuromorphic tasks and scalability by compressing SEW-ResNet 152 on the ImageNet benchmark.

## 2 PRELIMINARY

### 2.1 LEAKY-INTEGRATE-AND-FIRE (LIF) NEURON

We use the discrete LIF neuron model described in (Fang et al., 2021), with $V_{reset} = 0$. As summarized in Eqs. 1–3, $U[t]$ and $V[t]$ represent the neuron membrane potential after neuron dynamics, and after the trigger of a spike at time t, respectively. $I[t]$ denotes the external input current. $S[t]$ denotes the output spike train. $\tau_m$ is the membrane time constant, which governs the rate of decay for the membrane potential.

$$U[t] = \left(1 - \frac{1}{\tau_m}\right) V[t-1] + \frac{1}{\tau_m} I[t] \tag{1}$$

$$S[t] = \Theta(U[t] - V_{th}) \tag{2}$$

$$V[t] = U[t](1 - S[t]) \tag{3}$$

## 2.2 OPTIMAL BRAIN SURGEON

Neural network compression methods that use second-order approximation are extensions of the OBS framework (Hassibi & Stork, 1992), which considers the problem of accurately compressing a well-trained dense neural network. We briefly review the OBS rules for selecting the next weight to compress $w_p$, and the optimal compensation $\delta_p$, given the expected Hessian $\boldsymbol{H}$ of loss $\mathcal{L}$:

$$w_p = argmin_{w_p} \frac{c_{w_p}^2}{[\boldsymbol{H}^{-1}]_{pp}}, \delta_p = -\frac{c_{w_p}}{[\boldsymbol{H}^{-1}]_{pp}} \cdot [\boldsymbol{H}^{-1}]_{:p} \tag{4}$$

Here, $c_{w_p}$ denotes the changes to weight p (for pruning $c_{w_p} = w_p$, for quantization $c_{w_p} = w_p - \hat{w}_p$ where $\hat{w}_p$ is the nearest quantized expression of $w_p$). $[\boldsymbol{H}^{-1}]_{pp}$ denotes the $p^{th}$ diagonal of inverse Hessian, $[\boldsymbol{H}^{-1}]_{:p}$ is its $p^{th}$ column.

### 2.2.1 OPTIMAL BRAIN COMPRESSION HESSIAN MATRIX UPDATE

As a weight is pruned, its Hessian $\boldsymbol{H'} = \boldsymbol{H_{-p}}$, where $\boldsymbol{H_{-p}}$ represents matrix H with row p and column p removed. However, $\boldsymbol{H_{-p}^{-1}} \neq (\boldsymbol{H^{-1}})_{-p}$, and calculating the inverse hessian of the remaining matrix at each pruning iteration can incur a high computational cost. Building upon OBS, Optimal Brain Compression (Frantar et al., 2023b) introduced a straightforward method to obtain the inverse of a hessian matrix without row p and column p, using the Sherman-Morrison formula:

$$\boldsymbol{H_{-p}^{-1}} = (\boldsymbol{H^{-1}} - \frac{1}{[\boldsymbol{H^{-1}}]_{pp}} \boldsymbol{H_{:,p}^{-1}} \boldsymbol{H_{p,:}^{-1}})_{-p} \tag{5}$$

## 3 METHODOLOGY

**Module-wise compression** Since small current errors accumulate through LIF dynamics, we measure distortion at the output spike train. To achieve this, we compress SNNs *module-wise*: Every module comprises Linear/Conv(+BatchNorm) → LIF layers; BatchNorm and Conv can be folded into a single linear map, so each module becomes Linear(W) → LIF. (Figure 1 visualizes the module concept in an SNN). For an input spike tensor $X \in \{0, 1\}^{T \times d_{in}}$ and weights $W \in \mathbb{R}^{d_{in} \times d_{out}}$, the module produces a spike train $S = f(X, W) \in \{0, 1\}^{T \times d_{out}}$. We treat each module as an independent compression problem.

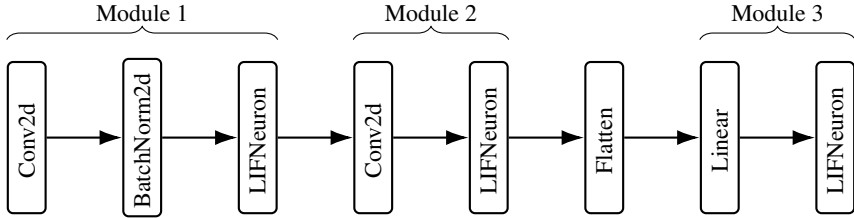

Figure 1: Example modules in a sample SNN. Each module ends with a LIF neuron layer and includes the linear, convolution, or batch normalization layers before it.

**Problem setup** The goal of module-wise compression is to find a "compressed" $W$, which we denote $\hat{W}$, that performs as similarly to the original weights as possible. More formally, the compressed weights $\hat{W}$ should minimize some loss function $\mathcal{L}$ while satisfying some constraints on $\hat{W}$, expressed as $\mathcal{C}(\hat{W}) > C$:

$$argmin_{\hat{W}} E_X[\mathcal{L}(f(X, W), f(X, \hat{W}))], \text{ s.t. } \mathcal{C}(\hat{W}) > C \tag{6}$$

The expectation is approximated using $N$ calibration samples.

### 3.1 SPIKING BRAIN COMPRESSION (SBC)

This paper proposes a framework for identifying an appropriate loss function $\mathcal{L}$ and obtaining the loss function's Hessian $\boldsymbol{H}$ for general SNNs. We refer to this framework as Spiking Brain Compression.

#### 3.1.1 LOSS FUNCTION $\mathcal{L}$

Let $\hat{S} = f(X, \hat{W})$, and $S = f(X, W)$. Due to the discrete nature of $S$ and $\hat{S}$, using the squared L2 Norm on $S - \hat{S}$ ignores the time distances between modified and original spikes. To address this, we used the squared VRD between the pre- and post-compression output spike trains as the loss function. Here we represent the VRD decaying exponential kernel $k(t)$ as a function of time constant $\tau$:

$$k[t] = \begin{cases} (1 - \frac{1}{\tau_m})^t \frac{1}{\tau_m} & \text{if } t \geq 0 \\ 0 & \text{if } t < 0 \end{cases} \tag{7}$$

The loss can then be expressed as:

$$\mathcal{L}(W) = VRD(S, \hat{S}) = ||S(t) * k(t) - \hat{S}(t) * k(t)||_2^2 = ||MS - M\hat{S}||_2^2 \tag{8}$$

Where the square matrix $M$ of size $d_{time} \times d_{time}$ denotes the convolution matrix of kernel $k(t)$ over $d_{time}$ timesteps. Since the output spike train of an LIF neuron depends solely on its own external input current, and $k(t)$ operates as a convolution along the time dimension, the VRD of each LIF neuron only depends on the weights feeding into the neuron. Consequently, the loss in Eq. 8 can be expressed as the sum of the square VRD of each neuron in the layer.

$$||MS - M\hat{S}||_2^2 = \sum_{j=1}^{d_{out}} ||MS_{:,j} - M\hat{S}_{:,j}||_2^2 \tag{9}$$

Therefore, the Hessian for the synaptic connections of each LIF neuron (i.e., each column of the weight matrix $W$) can be computed independently, as there are no inter-neuron dependencies. In the following section, we analyze the loss function of an individual neuron, with lower case $s$, $\hat{s}$, and $w$ representing the spiketrains and synaptic connections of an individual LIF neuron: $\mathcal{L} = ||Ms - M\hat{s}||_2^2 = ||Mf(X, w) - Mf(X, \hat{w})||_2^2$.

#### 3.1.2 SURROGATE MEMBRANE POTENTIAL (SMP)

Here, we provide SMP, an approximation of the exact Hessian that is computationally efficient and performs well empirically across both small and large SNNs. We first provide the Hessian $\mathcal{H}$ of the loss function $\mathcal{L}$ under the OBS framework. Detailed derivation can be found in Appendix A:

$$\mathcal{H}_{ij} = 2(M \frac{\partial s}{\partial u} x_j)^T M \frac{\partial s}{\partial u} x_i + 2(Ms - M\hat{s})^T M x_i \frac{\partial^2 s}{\partial u^2} x_j \tag{10}$$

Where $s$ and $u$ stand for the layer spike train and layer exact membrane potential, and $\frac{\partial s}{\partial u}$ is the $d_{times} \times d_{times}$ Jacobian, which we name $h'$, and $\frac{\partial^2 s}{\partial u^2}$ we rename $h''$.

We observe that the spike at time t $s[t] = \Theta(u[t] - V_{th})$ (Eq. 2) only depends on the membrane potential at time t. This means $h'$ and $h''$ are diagonal matrices. Inspired by surrogate gradient (Neftci et al., 2019), we replace the non-differentiable Heaviside function $\Theta$ with a differentiable gradient function $g$, thus $h'$ is a diagonal matrix where $h'_{ii} = g(u_i)$.

There are many different ways to design a surrogate gradient function $g$, and we intend to explore them in future work. Currently, we found that a constant function $g(u) = c$, works well for layer-wise compression. $g$ can be seen as a *shallow rectangle surrogate gradient* where every $u$ falls in the

active window. Since the OBS framework (Hassibi & Stork, 1992) works with the relative magnitude of the Hessian matrix, c cancels out. Without loss of generality, we set $c = 1$. Let $\boldsymbol{H} = E_X[H]$ be the expectation of the Hessian over a distribution of input X. With a constant function $g$, we also obtain $h'' = g' = 0$. We now obtain the Hessian for SMP:

$$\mathcal{H}_{\mathcal{SMT}ij} = 2(Mh'x_j)^T Mh'x_i = 2(MIx_j)^T MIx_i \tag{11}$$

$$\boldsymbol{H}_{SMT} = E_X[\mathcal{H}_{SMT}] = E_X[2(MIX)^T MIX] = E_X[2(MX)^T MX] \tag{12}$$

In fact, $\boldsymbol{H}_{SMT}$ is the exact Hessian of the least square form $||MXw - MX\hat{w}||_2^2$, which is the square L2 Loss on the membrane potential of a spiking neuron with the input current $Xw$ in the absence of spikes.

### 3.1.3 THE SBC PRUNING ALGORITHM

Here, we introduce the module-wise unstructured pruning framework for SNNs, utilizing SMP. We start by obtaining a per-module pruning target with Layer-Adaptive sparsity for the Magnitude-based Pruning Score (LAMPS) (Lee et al., 2021) to determine per-module pruning percentage from a global pruning target. Appendix C.3 provides a detailed explanation of LAMPS. LAMPS takes $O(d \cdot log(d))$, where $d$ represents the number of trainable parameters in the SNN.

**Step 1: Weight ordering.** To efficiently determine the pruning order of weights in each module, we record each weight's loss by performing OBS per neuron with a small batch size $B_{in}$. Each iteration, we take weights with $B_{in}$ smallest loss according to Eq. 4 and prune them together, then update the inverse Hessian with Woodbury Matrix Identity (Hassibi & Stork, 1992). The per-neuron full algorithm is given in Algorithm 1. For small $B_{in}$, this takes $O(\frac{d_{in}^3}{B_{in}})$ time and $O(d_{in})$ space.

In actuality, we can batch $B_{out}$ neurons in parallel, given the available GPU resources. This produces a loss for each weight in the module, which we then sort and create a mask $\mathbf{M}$ of pruned weights according to the module pruning target. This takes $O(\frac{d_{out}}{B_{out}} \cdot \frac{d_{in}^3}{B_{in}})$ time and $O(B_{out} \cdot d_{in}^2)$ space.

**Step 2: Weight pruning.** With a mask $\mathbf{M}$, we can directly update weights of a neuron according to its local set of weights to remove $\mathbb{P} = \mathbf{M}_{:,i}$ via the group OBS formula $\delta_{\mathbb{P}} = -\mathbf{H}_{:,\mathbb{P}}^{-1}((\mathbf{H}^{-1})_{\mathbb{P}})^{-1}\mathbf{W}_{:,i}$. This takes $O(\frac{d_{out}}{B_{out}} \cdot d_{in}^3)$ time and $O(B_{out} \cdot d_{in}^2)$ space, but it is generally faster than the weight ordering step.

**Complexity.** To summarise, per-module SBC has space complexity $O(B_{out} \cdot d_{in}^2)$ and time complexity $O(\frac{d_{out}}{B_{out}} \cdot \frac{d_{in}^3}{B_{in}})$, with $d_{in}, d_{out}$ refer to the input and output dimension of the linearized pa-

---

**Algorithm 1** Losses $\mathbf{L}$ for weights $\mathbf{w}$ of neuron with $\mathbf{H}^{-1} = (2(\mathbf{MX})^\top \mathbf{MX})^{-1}$, in $O(\frac{d_{in}^3}{B_{in}})$ time.

> $\mathbb{M} \leftarrow \{1, \ldots, d_{in}\}$
> $\mathbf{L} \leftarrow \{\}$
> **for** $i = 1, 1 + B_{in}, \ldots, d_{in}$ **do**
>      $s_p \leftarrow \dfrac{w_p^2}{[\mathbf{H}^{-1}]_{pp}} \; \forall p \in \mathbb{M}$
>      $\mathbb{P} \leftarrow$ indices of the $B_{in}$ smallest $\{s_p\}_{p \in \mathbb{M}}$
>      $\mathbf{L}[p] \leftarrow s_p \; \forall p \in \mathbb{P}$
>      $\mathbf{w}_{\mathbb{P}} \leftarrow \mathbf{w}_p \; \forall p \in \mathbb{P}$
>      $\mathbf{w} \leftarrow \mathbf{w} - \mathbf{H}_{:,\mathbb{P}}^{-1}((\mathbf{H}^{-1})_{\mathbb{P}})^{-1} \cdot \mathbf{w}_{\mathbb{P}}$
>      $\mathbf{H}^{-1} \leftarrow \mathbf{H}^{-1} - \mathbf{H}_{:,\mathbb{P}}^{-1}((\mathbf{H}^{-1})_{\mathbb{P}})^{-1}\mathbf{H}_{\mathbb{P},:}^{-1}$
>      $\mathbb{M} \leftarrow \mathbb{M} \setminus \{\mathbb{P}\}$
> **end for**

**Algorithm 2** SBC Pruning, with SNN $\mathcal{M}$, calibration data $\mathcal{X}$, model sparsity target $s \in [0, 1]$

> $\mathbb{M} \leftarrow$ modules($\mathcal{M}$)      ▷ get set of modules
> **for all** $\mathbf{m} \in \mathbb{M}$ **do**
>      $s_m =$ LAMPS($\mathcal{M}, s, \mathbf{m}$)   ▷ module target
>      $\mathbb{X} \leftarrow$ module m input data from $\mathcal{X}, \mathcal{M}$
>      $\mathbf{H} = \frac{2}{|\mathbb{X}|} \sum_{X \in \mathbb{X}} (MX)^T MX$
>      $\mathbf{W} \leftarrow$ weights of $\mathbf{M}$
>      $\mathbf{L} \leftarrow$ **Algorithm 1**($\mathbf{W}, \mathbf{H}^{-1}$)
>      $\mathbf{M} \leftarrow$ indicies of $|\mathbf{W}| \cdot s_\mathbf{m}$ smallest $\mathbf{L}$
>      **for** $i = 1, \ldots d_{out}$ **do**
>          $\mathbb{P} \leftarrow \mathbf{M}_{:,i}$
>          $\delta_{\mathbb{P}} \leftarrow -\mathbf{H}_{:,\mathbb{P}}^{-1}((\mathbf{H}^{-1})_{\mathbb{P}})^{-1}\mathbf{W}_{:,i}$
>          $\mathbf{W}_{:,i} \leftarrow \mathbf{W}_{:,i} + \delta_{\mathbb{P}}$
>      **end for**
> **end for**

rameterized layer in the module. Batch sizes $B_{in}$ and $B_{out}$ are adjusted to balance time and space according to hardware availability.

**Deep SNNs.** SBC can also handle compression of deep SNNs like various Spiking ResNets (Fang et al., 2022) and Spiking Transformer (Yao et al., 2023; Zhou et al., 2022) architectures. Appendix B provides implementation details. The intuition is to treat each module separately, and when two different parameterized layers feed into the same spiking neuron layer, to concatenate the parameterized layers' weights and inputs.

### 3.1.4 THE SBC QUANTIZATION ALGORITHM

The SBC quantization algorithm largely follows GPTQ (Frantar et al., 2023a), replacing the layer-wise Hessian with $\mathbf{H}_{SMP}$. It has per-module space complexity $O(d_{in}^2)$ and time complexity $O(max\{d_{out} \cdot d_{in}^2, d_{in}^3\})$.

### 3.2 COMPARING SBC WITH OBC

A naive implementation of OBC on SNN would have a loss function directly on the output of the linear layer. It has the Hessian:

$$\boldsymbol{H}_{OBC} = E_X[\mathcal{H}_{OBC}] = E_X[2X^T X] \tag{13}$$

Recall Eq.11, $M\frac{\partial s}{\partial u} = Mh'$ always preserves the lower triangle structure (for spike trains with spikes) because $M$ is lower triangular, and $h'$ is diagonal. This means the simplification $Mh' = I$, i.e. $\boldsymbol{H}_{OBC}$, is typically weaker than keeping $M$, i.e. $\boldsymbol{H}_{SMT}$.

## 4 EXPERIMENTS

To explore how SBC works in practice, we performed pruning and quantization on models trained with neuromorphic and static datasets. SNNs can leverage unstructured pruning to reduce compute (Shi et al., 2024) on neuromorphic hardware, unlike ANNs on GPUs. For this reason, the experiments are focused on pruning, while we conducted quantization experiments only on N-MNIST, CIFAR10-DVS, and DVS128-Gesture.

The specifications of each model are listed in Table 1. SNN implementation is based on the Spiking-Jelly library (Fang et al., 2023). Experimented were conducted on a single Nvidia A4500 (20GB) GPU.

Table 1: Datasets and SNN Architectures used for Compressions. $\tau_m$ is the membrane time constant of the LIF neuron layers, and T is the number of timesteps. Note that the SEW-ResNet family uses Integrate-and-Fire (IF) neurons, which are equivalent to LIF neurons with $\tau_m = \infty$.

| Dataset | Architecture | Input Shape | $\tau_m$ | T | Accuracy(%) |
|---------|--------------|-------------|----------|-----|-------------|
| N-MNIST | 2FC | 34x34x2 | 2.0 | 100 | 98.31 |
| CIFAR10-DVS | 4Conv+2FC | 64x64x2 | 2.0 | 20 | 71.50 |
| DVS128-Gesture | 5Conv+2FC | 128x128x2 | 2.0 | 20 | 95.14 |
| ASL-DVS | 4Conv+1FC | 180x240x2 | 2.0 | 30 | 96.53 |
| CIFAR100 | VGG16SNN | 32x32x3 | 1.33 | 5 | 71.05 |
| ImageNet | SEW-ResNet Family | 256x256x3 | $+\infty$ | 4 | 63.12-69.18 |

Four models were trained on neuromorphic datasets N-MNIST, DVS128 Gesture, DVS-CIFAR10, and ASL-DVS. N-MNIST was trained on the Adam optimizer with $lr = 0.001$ and max epoch 200. DVS-CIFAR10 and DVS128-Gestures were trained on the Adam optimizer with Cosine Annealing $lr = 0.001$, and max epoch 512. ASL-DVS was trained on SGD optimizer with Cosine Annealing $lr = 0.001$ for 90 epochs. N-MNIST, DVS128-Gesture, and ASL-DVS maintained their original size, whereas DVS-CIFAR10 was downsized due to the long training time.

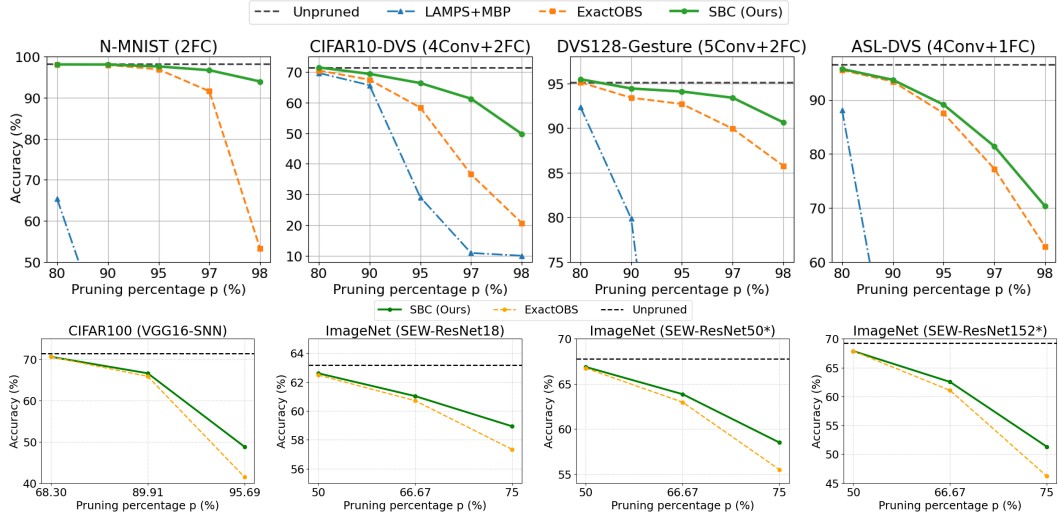

Figure 2: Neuromorphic (top row) and static (bottom row) dataset pruning results, post-training one-shot SNN pruning algorithms. (*) models are largest SNNs pruned to date. Tables: Appx. D

The Spiking VGG16 model was chosen for CIFAR-100 dataset. It was trained on SGD optimizer for 300 epochs, with steps at 150 and 225, using an initial learning rate of $lr_{initial} = 0.1$. SEW-ResNet family models checkpoints were retrieved from its GitHub repository.

### 4.1 PRUNING RESULT AND ANALYSIS

#### 4.1.1 SBC VS ONE-SHOT POST-TRAINING SNN PRUNING ALGORITHMS

We compared the performance of the naive magnitude-based pruning (MBP) shown in LAMPS, the naive ExactOBS (OBC's pruning algorithm) implementation on SNN, and SBC at different sparsity levels across all datasets. The accuracies are recorded in Fig. 2. Note that all three post-training one-shot methods pruned the same proportion of weights from each module according to LAMPS; the difference is in how each module prunes its weights.

Across neuromorphic benchmarks SBC matches baseline accuracy at p=0 and substantially outperforms both ExactOBS and LAMPS+MBP as sparsity increases. In particular, while all methods start from the same unpruned accuracy, SBC maintains high accuracy under extreme pruning (e.g. N-MNIST: -1.59% at 97% sparsity compared to ExactOBS's -45.07%; DVS128: -1.74% at 97% sparsity compared to ExactOBS's -9.38%), and shows similar robustness on CIFAR10-DVS and ASL-DVS where competing methods degrade more rapidly. In short, SBC preserves performance at low sparsity and extends the usable sparsity range relative to other one-shot post-training methods.

The static-dataset results mirror this trend: SBC has the largest gains compared to ExactOBS at very high sparsities (e.g. CIFAR-100 VGG16-SNN: +7.47% at 75% sparsity). On ImageNet across SEW-ResNet variants, SBC consistently matches or improves ExactOBS, demonstrating that SBC is the SOTA one-shot post-training pruning algorithm for SNNs.

#### 4.1.2 SBC VS PAT ALGORITHMS

We compare SBC (with and without post-pruning fine-tuning) against representative PAT baselines including LTH (Kim et al., 2022), UPF (Shi et al., 2024), and STDS (Chen et al., 2022); accuracy and wall-clock pruning times are reported in Fig. 3. Across CIFAR-100 and ImageNet, SBC reaches competitive accuracy, and adding a short fine-tuning phase further closes the gap to PAT methods while incurring minimal additional computational cost.

Concretely, on CIFAR-100 at 68.30% and 89.91% sparsity, SBC improves top-1 accuracy by 2.02% and 0.06% versus the IMP found in LTH, while reducing pruning time by roughly two orders of magnitude. SBC also outperforms LTH Early Bird methods across the evaluated sparsity targets

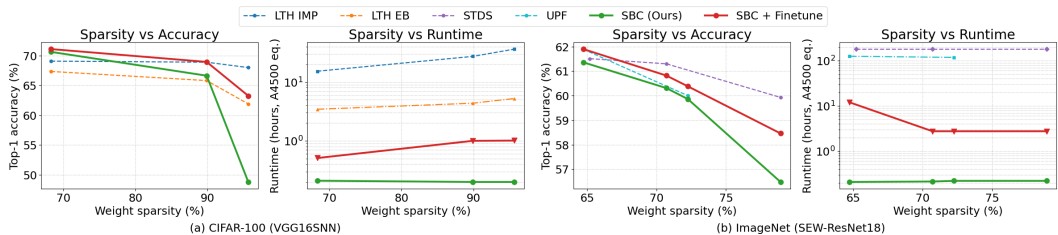

Figure 3: Large static dataset pruning results, SBC (Ours) vs PAT methods. Tables: Appx. D

while running 5–7× faster. On ImageNet, SBC matches or approaches UPF/STDS accuracy but requires substantially less time (three orders of magnitude faster for the pruning step); with fine-tuning, SBC surpasses UPF's accuracy but remains slightly below STDS. These results indicate that SBC is a cost-effective alternative to full PAT pipelines: it delivers strong, deployable sparsity with far lower compute and provides a flexibility to trade a small additional cost for further accuracy gains.

### 4.1.3 LARGE SNNS PRUNING

Crucially, the lower computation cost enables SBC to prune large SNNs such as SEW ResNet50/152, when PAT methods become too costly. To the best of our knowledge (Kim et al., 2022; Chen et al., 2022; Shi et al., 2024), this experiment is the first time SNNs with more than 34 layers have been pruned, and the first SNNs trained on an ImageNet-scale dataset with more than 19 layers have been pruned (Kim's experiment on Spiking ResNet-34 (Kim et al., 2022) was trained on Tiny-ImageNet (Le & Yang, 2015)). To the best of our knowledge, SEW-ResNet152 is also the largest and deepest SNN model that has been submitted to unstructured pruning by any algorithm to date. This reaffirms SBC as the SOTA one-shot post-training compression method for SNNs.

In conclusion, the SBC pruning method consistently outperforms current one-shot post-training pruning algorithms for SNNs, achieving one-shot post-training pruning SOTA for SNNs. It is competitive and sometimes even exceeds the accuracy of the SOTA SNN iterative pruning method (Kim et al., 2022), with 1 to 3 orders of magnitude of time savings.

### 4.2 QUANTIZATION RESULT AND ANALYSIS

We compared SBC's quantization accuracy with a baseline method and vanilla implementation of GPTQ (Frantar et al., 2023a), the state-of-the-art quantization derivative of OBC, on SNN. The quantization experiment used the same trained models and sample data as the pruning experiment. The models were quantized to 4, 3, and 2 bit widths.

This work used the GPTQ framework for quantization experiments. GPTQ quantizes each linear/convolution ANN layer according to standard uniform per-channel symmetric quantization on the min-max grid, similar to (Dettmers et al., 2022). We chose a symmetric quantization grid for the low computing cost when deploying to neuromorphic hardware. The weights are then quantized in the ascending diagonal inverse Hessian order, which results in a quantization order from biggest to smallest loss in an equal spacing quantization grid according to Eq. 4. The quantization grids and weight order are the same as GPTQ. It is notable that SBC quantization can be applied to arbitrary quantization grids.

The baseline method, which **R**ounds each weight **T**o its **N**earest quantized target value, is called RTN. Note that all three methods (RTN, vanilla GPTQ, and SBC) share the same quantization grid, which means all three methods have precisely the same set of quantization target values to round to, and the difference in accuracy only comes from the choice of target. The result of the quantization experiment can be found in Table 2. Each quantization result is taken as the average of 5 runs with randomly chosen sample datasets.

SBC's performance is generally better than that of RTN and GPTQ. It can quantize the CIFAR10-DVS model to 3-bit precision with a 1.86% drop in accuracy. However, SBC does not consistently outperform RTN and GPTQ in higher-precision quantization for CIFAR10-DVS and DVS128-

Table 2: Neuromorphic datasets quantization result

| Setting | Method | Accuracy(%) | | | |
|---|---|---|---|---|---|
| | | $q = 32bit$ | $4bit$ | $3bit$ | $2bit$ |
| N-MNIST 2FC | RTN | 98.31 | 97.58 | 62.26 | 20.34 |
| | GPTQ | 98.31 | **98.16** | 97.33 | 64.29 |
| | **SBC(Ours)** | 98.31 | 98.14 | **97.81** | **92.40** |
| CIFAR10-DVS 4Conv+2FC | RTN | 71.50 | 66.40 | 52.70 | 20.50 |
| | GPTQ | 71.50 | **70.30** | 67.56 | 59.40 |
| | **SBC(Ours)** | 71.50 | 69.96 | **69.64** | **60.70** |
| DVS128-Gesture 5Conv+2FC | RTN | 95.14 | 87.15 | 78.47 | 53.82 |
| | GPTQ | 95.14 | 86.67 | 81.25 | 62.78 |
| | **SBC(Ours)** | 95.14 | **87.43** | **81.88** | **64.79** |

Gesture. This is caused by the weight compensation that pushes latter quantized weights to more extreme values. These extreme values are outside of the range of the quantization grid. Nevertheless, SBC achieves the best quantization result out of the three methods at the lowest bit representation tested (2bit), outperforming the next best method by 28.11%, 1.30%, and 2.01%, respectively.

## 5 LIMITATIONS AND FUTURE WORKS

In quantization experiments, this work observed the out-of-bounds problem with the min-max quantization grid, and a more rigorous study of the relationship between the choice of the quantization grid and quantization performance is needed.

Future works include exploring more sophisticated surrogate gradients $g'$, $MJ$, or even a general $d_{time} \times d_{times}$ matrix in place of $M$ in the current $\boldsymbol{H}_{SMP}$. These matrices can be adjusted depending on each neuron's properties that are not expected to change during compression, such as spike rates. However, we would like to note that a naive implementation of custom Hessian matrices for each output would have high space complexity $\Theta(d_{out} \cdot d_{in}^2)$. In the conv2 layer of the last blocks in the SEW-ResNet152, which has linearized $d_{in} = 4608$ and $d_{out} = 512$, the Hessian would take up $512 \times 4608^2 \times 4bit = 43.5GB$ of space at fp32. As SNN architectures scale, this cost grows quickly, making a lower-complexity strategy essential for practical compression use cases.

On the experimental front, we plan to extend our study to deeper and larger SNNs, such as Spikformers (Zhou et al., 2024) and SpikeGPT (Zhu et al., 2024). Beyond pruning alone, we will explore joint compression schemes that combine pruning and quantization for SNNs. Finally, we aim to deploy the compressed models on neuromorphic hardware, e.g., Loihi 2 (Orchard et al., 2021), to directly measure real-world compute and power efficiency gains.

## 6 CONCLUSION

In conclusion, this paper proposes a one-shot, efficient, post-training compression framework for SNNs, utilizing a second-order approximation of the per-layer spike train loss to dynamically compress and compensate for the compression. Through empirical analysis, this paper shows SBC's effectiveness in compressing SNNs trained for neuromorphic and static datasets, outperforming current SOTA on SNN post-training one-shot methods, and competitive with the accuracy of iterative retraining compression methods while providing 1-2 orders of magnitude decrease in compression time. We expect this work to pave the way for the efficient compression of very large SNNs, such as deep SNNs and Spiking Transformers, for low-power, edge computing, and robotic use cases.

## REPRODUCIBILITY STATEMENT

We provide anonymized code and configs in the supplemental materials, covering training, pruning, and evaluation. Detailed derivations are included in Appendix A and B. Dataset preprocessing steps and hyperparameters are described in Table 1; architecture and implementation details are in Appendix C. All graphs in the main paper has numerical results presented as tables in Appendix D

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

## A    DERIVING EXACT HESSIAN FOR INDIVIDUAL LIF NEURON

We derive the first-order derivative of loss function $\mathcal{L}$ against weight $w_i$:

$$
\begin{aligned}
\frac{\partial \mathcal{L}}{\partial w_i} &= \frac{\partial \mathcal{L}}{\partial s}\frac{\partial s}{\partial w_i} = \frac{||Ms - M\hat{s}||_2^2}{\partial s}\frac{\partial s}{\partial w_i} \\
&= 2(Ms - M\hat{s})^T M \frac{\partial s}{\partial w_i}
\end{aligned}
\tag{14}
$$

As is customary in OBS framework, with a well-trained neural network, we assume $\nabla_{w_j}\mathcal{L} = 0$ for all weights $w_i$. Now we can get the second order derivative:

$$
\begin{aligned}
\mathcal{H}_{ij} &= \frac{\partial^2 \mathcal{L}}{\partial w_i \partial w_j} \\
&= \frac{\partial}{\partial w_j}(2(Ms - M\hat{s})^T M \frac{\partial s}{\partial w_i}) \quad \text{(by equation 14)} \\
&= 2[\frac{\partial}{\partial w_j}(a^T b)] \quad \text{where } a = (Ms - M\hat{s}), b = M\frac{\partial s}{\partial w_i} \\
&= 2[(\frac{\partial a}{\partial w_j})^T b + a^T(\frac{\partial b}{\partial w_j})] \quad \text{(product rule)} \\
&= 2[(\frac{\partial(Ms - M\hat{s})}{\partial w_j})^T M \frac{\partial s}{\partial w_j} + (Ms - M\hat{s})^T \frac{\partial^2 Ms}{\partial w_i \partial w_j}] \quad \text{expand a and b} \\
&= 2[(\frac{\partial(Ms)}{\partial w_j})^T M \frac{\partial s}{\partial w_j} + (Ms - M\hat{s})^T M \frac{\partial^2 s}{\partial w_i \partial w_j}] \\
&= 2(M\frac{\partial s}{\partial w_j})^T M \frac{\partial s}{\partial w_i} + 2(Ms - M\hat{s})^T M \frac{\partial^2 s}{\partial w_i \partial w_j}
\end{aligned}
\tag{15}
$$

We can further derive $\frac{\partial s}{\partial w_i}$ and $\frac{\partial^2 s}{\partial w_i \partial w_j}$:

$$\frac{\partial s}{\partial w_i} = \frac{\partial s}{\partial u}\frac{\partial u}{\partial w_i} = \frac{\partial s}{\partial u}x_i$$

$$\frac{\partial^2 s}{\partial w_i \partial w_j} = \frac{\partial}{\partial w_j}(\frac{\partial s}{\partial u}x_i) = x_i\frac{\partial}{\partial w_j}(\frac{\partial s}{\partial u}) = x_i[\frac{\partial}{\partial u}(\frac{\partial s}{\partial u})]\frac{\partial u}{\partial w_j} = x_i\frac{\partial^2 s}{\partial u^2}x_j \tag{16}$$

Where $x_i, x_j$ denotes time-varying pre-synaptic spike trains to synaptic connections $w_i$ and $w_j$, i.e. $x_i = X_{:,i}$. We can now further simplify the exact Hessian Eq. 15:

$$\mathcal{H}_{ij} = 2(M\frac{\partial s}{\partial u}x_j)^T M\frac{\partial s}{\partial u}x_i + 2(Ms - M\hat{s})^T Mx_i\frac{\partial^2 s}{\partial u^2}x_j \tag{17}$$

## B  DERIVATION ON HOW TO APPLY SBC TO VARIOUS SPIKING RESNETS AND SPIKING TRANSFORMER ARCHITECTURES

### B.1  SPIKING RESNETS

**Shortcut handling in Spiking ResNets**  We derive the SBC loss with surrogate membrane potential (SMP) for two popular Spiking ResNet variants: Spiking-Element-Wise (SEW) ResNet Fang et al. (2022) and SpikingResNet Hu et al. (2023). In SEW ResNet, the shortcut follows the spiking layer; from SBC's perspective, the two branches are already independent modules and can be pruned separately.

In SpikingResNet, the shortcut precedes the spiking layer. Thus the external current to the next spiking layer is the sum of the convolution output and the shortcut output. Consider one neuron with linearized convolution output $X_1w_1$, shortcut output $X_2w_2$, and surrogate membrane-potential operator $M$. The shortcut loss is

$$L_{\text{shortcut}} = \mathbb{E}_X\Big[\|M(X_1w_1 + X_2w_2) - M(X_1\hat{w}_1 + X_2\hat{w}_2)\|_2^2\Big] \tag{18}$$

$$= \mathbb{E}_X\Bigg[\bigg\|M\bigg([\,X_1\ \ X_2\,]\begin{bmatrix}w_1\\w_2\end{bmatrix}\bigg) - M\bigg([\,X_1\ \ X_2\,]\begin{bmatrix}\hat{w}_1\\\hat{w}_2\end{bmatrix}\bigg)\bigg\|_2^2\Bigg]. \tag{19}$$

Hence we may concatenate the inputs and weights of the last convolution and the shortcut in each SpikingResNet block, forming a new module with input $X = [\,X_1\ \ X_2\,]$ and weights $W = \begin{bmatrix}w_1\\w_2\end{bmatrix}$. Compression is then performed on this module. If the shortcut has no trainable parameters, we simply mask out the weights associated with $X_2$ before compression. *Notation:* $[\,X_1\ \ X_2\,]$ denotes horizontal concatenation; $\begin{bmatrix}w_1\\w_2\end{bmatrix}$ denotes vertical concatenation.

### B.2  SPIKING TRANSFORMERS

This section examines the applicability of **SBC** to popular *spiking–transformer* architectures, and found that it extends cleanly to Spikformer Li et al. (2022), Spikformer V2 Zhou et al. (2024), and the Spike-Driven Transformer (SDT) Yao et al. (2023).

**Spikformer family.**  In Spikformer, the weights of the Spiking Self-Attention (SSA) layers $(W_Q, W_K, W_V)$ belong to a Linear $\rightarrow$ BN $\rightarrow$ Spiking block and can therefore be treated as *three* SBC modules. Each MLP block follows Linear$\rightarrow$BN$\rightarrow$Spiking$\rightarrow$Linear$\rightarrow$BN$\rightarrow$Spiking, giving *two* additional modules. Spikformer V1 uses Spiking Patch Splitting (SPS), whereas V2 employs a Spiking Convolution Stem (SCS); both are sequential convolutions that fit the SBC module-wise compression definition in Section 3.

**Spike-Driven Transformer (SDT).**  In SDT, the shortcut precedes the spiking layer, so the external current is the sum of the previous BN output and the shortcut. Let the main-path output be $XW$,

the shortcut output $U$, and $M$ the surrogate membrane-potential operator. For a single spiking layer, the shortcut loss is

$$
\begin{aligned}
L_{\text{shortcut}} &= \mathbb{E}_X\big[\|M(XW + U) - M(X\hat{W} + U)\|_2^2\big] \\
&= \mathbb{E}_X\big[\|MXW + MU - MX\hat{W} - MU\|_2^2\big] \\
&= \mathbb{E}_X\big[\|MXW - MX\hat{W}\|_2^2\big].
\end{aligned}
\tag{20}
$$

Hence, the shortcut can be ignored during compression, allowing all SDT modules to be treated exactly as in Spikformer.

## C EXPERIMENT IMPLEMENTATION DETAILS

Note that we provide anonymised code for all experiments.

### C.1 STATIC MODELS

We use the Spiking VGG16 model implemented by Kim et al. (2022) for CIFAR100 experiments. We used pre-trained SEW-ResNet family of models for our ImageNet pruning experiments.

### C.2 NEUROMORPHIC MODELS

The detailed architectures of the models applied to each neuromorphic dataset are given in the figures 4, 6, 5, 7. All LIF neuron layers use the surrogate function $Atan()$, time constant $\tau = 2.0$. Note the downsizing of CIFAR10-DVS dataset to 64x64, due to the computation limitation of our hardware.

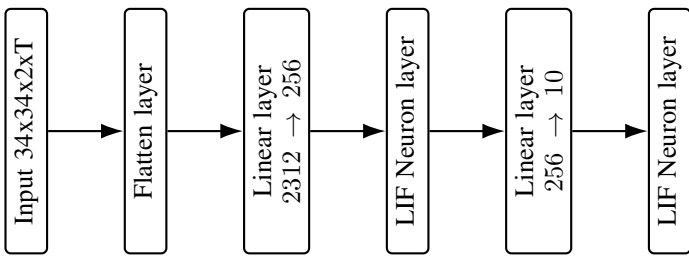

Figure 4: N-MNIST 2FC architecture

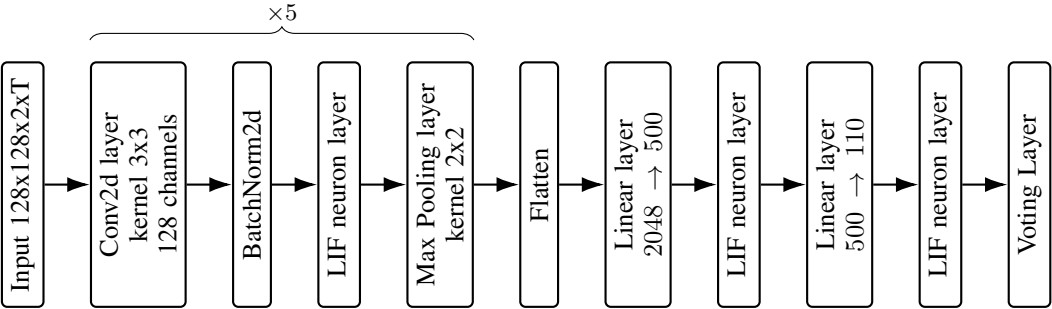

Figure 5: DVS128-Gesture 5Conv-2FC architecture

### C.3 PRUNING EXPERIMENTS DETAILS

The pruning percentage for each prunable layers in the module is pre-determined with LAMPS Lee et al. (2021). Here is a summary:

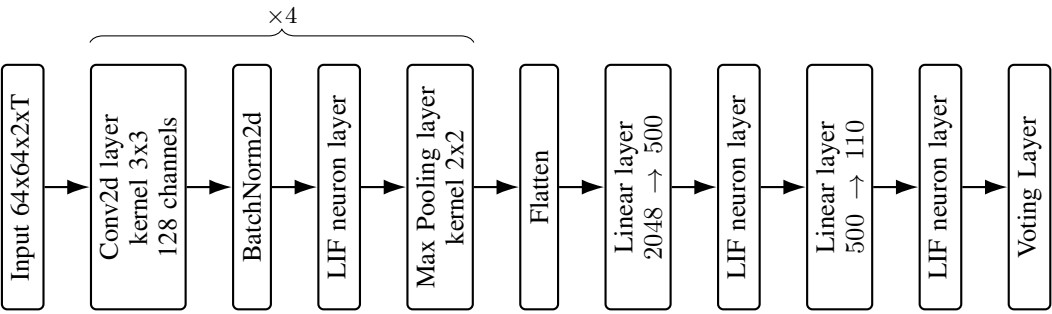

Figure 6: CIFAR10-DVS 4Conv-2FC architecture

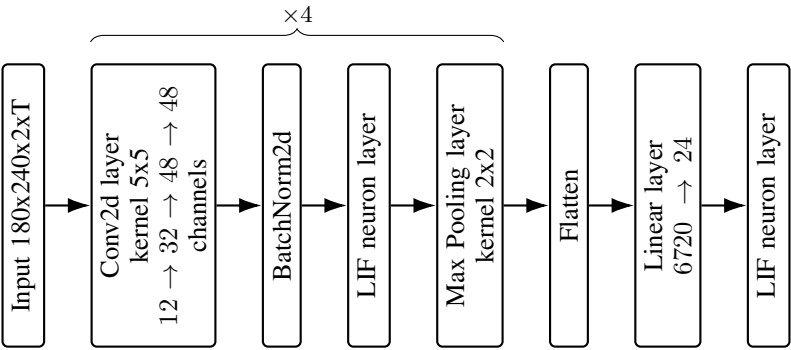

Figure 7: ASL-DVS 4Conv-1FC architecture

**LAMPS** Let the weights of layer $l$ be flattened into $\mathbf{W}^{(l)} \in \mathbb{R}^{N_l}$, and let $\pi_l$ be the permutation that orders them in non-decreasing magnitude:

$$\left|W^{(l)}_{\pi_l(1)}\right| \leq \left|W^{(l)}_{\pi_l(2)}\right| \leq \cdots \leq \left|W^{(l)}_{\pi_l(N_l)}\right|.$$

Then for $u = 1, \ldots, N_l$, the *LAMP score* of the $u$-th smallest weight in layer $l$ is

$$\text{LAMP}\big(l, \pi_l(u)\big) = \frac{\big(W^{(l)}_{\pi_l(u)}\big)^2}{\sum_{v=u}^{N_l} \big(W^{(l)}_{\pi_l(v)}\big)^2}. \tag{21}$$

To prune to a global sparsity level $S$, collect all scores $\{\text{LAMP}(l, \pi_l(u))\}$ across layers, and remove $\lfloor S \sum_l N_l \rfloor$ weights from each layer.

The weights in each modules are then pruned in the order suggested by LAMPS scores, however, the exact weights pruned and compensations are determined by the SBC pruning algorithm. The exact implementation can be found in supplementary file osbc_prune.py

## C.4 QUANTIZATION

The quantization grid is chosen as a symmetric per-channel fixed-grid of width $\Delta_c$. This means the maximum absolute value of a channel of weights equals the maximum value of the quantization grid. The symmetric quantization grid enables easy scalar conversion between compressed and original quantized values.

$$\hat{w}_c = w_{c_{quantized}} \times \Delta_c, w_{c_{quantized}} \in \{2^{n-1}, ..., -1, 0, 1, ...2^{n-1} - 1\} \tag{22}$$

For an n-bit quantization grid for channel c. The implementation of this maximum value grid is lifted directly from GPTQ Frantar et al. (2023a) implementation on GitHub. It can be found in the supplementary file quantizer.py.

The quantization experiment was run five times at each quantization level tested, with five different seeds for PyTorch and NumPy random generators used to select the sample dataset from the training dataset. The exact seeds can be found in modelutils.py. The implementation for OBSC quant can be found in osbc_quant.py in the supplementary material.

# D    EXPERIMENT RESULT TABLES

Table 3: Neuromorphic dataset pruning result

| Setting | Method | $p = 0\%$ | 80.00% | 90.00% | 95.00% | 97.00% | 98.00% |
|---|---|---|---|---|---|---|---|
| | | | | Accuracy(%) | | | |
| N-MNIST 2FC | LAMPS+MBP | 98.31 | 65.39 | 29.40 | 35.13 | 26.94 | 25.60 |
| | ExactOBS | 98.31 | **98.16** | 98.01 | 96.91 | 91.59 | 53.24 |
| | **SBC(Ours)** | 98.31 | 98.13 | **98.10** | **97.63** | **96.72** | **93.97** |
| CIFAR10-DVS 4Conv+2FC | LAMPS+MBP | 71.50 | 69.70 | 65.70 | 29.10 | 10.90 | 10.00 |
| | ExactOBS | 71.50 | 70.50 | 67.50 | 58.40 | 36.60 | 20.70 |
| | **SBC(Ours)** | 71.50 | **71.50** | **69.40** | **66.40** | **61.30** | **49.80** |
| DVS128-Gesture 5Conv+2FC | LAMPS+MBP | 95.14 | 92.36 | 79.86 | 35.13 | 9.03 | 8.33 |
| | ExactOBS | 95.14 | 95.13 | 93.40 | 92.7 | 89.93 | 85.76 |
| | **SBC(Ours)** | 95.14 | **95.48** | **94.44** | **94.10** | **93.40** | **90.63** |
| ASL-DVS 4Conv+1FC | LAMPS+MBP | 96.53 | 88.11 | 36.48 | 11.76 | 7.19 | 6.08 |
| | ExactOBS | 96.53 | 95.51 | 93.41 | 87.52 | 77.23 | 62.78 |
| | **SBC(Ours)** | 96.53 | **95.73** | **93.72** | **89.12** | **81.40** | **70.30** |

Table 4: Static dataset pruning results: SBC vs. ExactOBS

| Dataset | Arch. | Time step | Top-1 Acc. **SBC(Ours)** (%) | Top-1 Acc. ExactOBS (%) | Weight Sparsity (%) | Time (h) A4500 Eq. | Calib. size (% train) |
|---|---|---|---|---|---|---|---|
| CIFAR100 | VGG16-SNN | 5 | 71.40 | 71.40 | 0 | - | - |
| | | | **70.63** | **70.63** | 68.30 | 0.21 | 20% |
| | | | **66.44** | 65.89 | 89.91 | 0.20 | 20% |
| | | | **48.93** | 41.46 | 95.69 | 0.20 | 20% |
| ImageNet | SEW-ResNet18 | 4 | 63.18 | 63.18 | 0 | - | - |
| | | | **62.61** | 62.49 | 50.00 | 0.208 | 2% |
| | | | **61.03** | 60.71 | 66.67 | 0.214 | 2% |
| | | | **58.94** | 57.34 | 75.00 | 0.220 | 2% |
| | SEW-ResNet50† | 4 | 67.78 | 67.78 | 0 | - | - |
| | | | **66.88** | 66.75 | 50.00 | 0.346 | 2% |
| | | | **63.86** | 62.97 | 66.67 | 0.344 | 2% |
| | | | **58.50** | 55.50 | 75.00 | 0.341 | 2% |
| | SEW-ResNet152† | 4 | 69.26 | 69.26 | 0 | - | - |
| | | | **67.88** | **67.88** | 50.00 | 0.783 | 2% |
| | | | **62.53** | 61.06 | 66.67 | 0.808 | 2% |
| | | | **51.30** | 46.19 | 75.00 | 0.723 | 2% |

[*] **Bolded** entries represent best accuracies in sparsity class
[†] Deepest SNN models pruned to date. No existing PAT methods have pruned them.

Table 5: Static dataset pruning results, SBC compared with Pruning-Aware-Training methods

| Dataset / Model | Time step | Pruning Method | Top-1 Acc. (%) | Weight Sparsity (%) | Time (h) A4500 Eq. | Calib. size (% train) | Finetune epochs |
|---|---|---|---|---|---|---|---|
| CIFAR100 VGG16-SNN | 5 | LTH IMP | 69.08 | 68.30 | 15.23 | – | – |
| | | | 68.90 | 89.91 | 27.22 | – | – |
| | | | **68.00** | 95.69 | 36.22 | – | – |
| | | LTH EB | 67.35 | 68.30 | 3.44 | – | – |
| | | | 65.82 | 89.91 | 4.36 | – | – |
| | | | 61.90 | 95.69 | 5.24 | – | – |
| | | **SBC (Ours)** | 70.63 | 68.30 | **0.21** | 20% | – |
| | | | 66.44 | 89.91 | **0.20** | 20% | – |
| | | | 48.93 | 95.69 | **0.20** | 20% | – |
| | | **SBC + Finetune** | **71.10(+0.47†)** | 68.30 | 0.51 | 20% | 25.0 |
| | | | **68.96(+2.52†)** | 89.91 | 0.81 | 20% | 50.0 |
| | | | 63.34(+14.41†) | 95.69 | 0.81 | 20% | 50.0 |
| ImageNet SEW-ResNet18 | 4 | STDS | 61.51 | 65.21 | 180 | – | – |
| | | | **61.30** | 70.71 | 180 | – | – |
| | | | **59.93** | 78.92 | 180 | – | – |
| | | UPF | 61.89 | 64.74 | 125 | – | – |
| | | | 60.00 | 72.26 | 118 | – | – |
| | | **SBC (Ours)** | 61.36 | 64.74 | **0.208** | 2% | – |
| | | | 60.31 | 70.71 | **0.213** | 2% | – |
| | | | 59.86 | 72.26 | **0.220** | 2% | – |
| | | | 56.48 | 78.92 | **0.220** | 2% | – |
| | | **SBC + Finetune** | **61.90(+0.54†)** | 64.74 | 12.00 | 2% | 5.0 |
| | | | 60.82(+0.51†) | 70.71 | 2.75 | 2% | 1.0 |
| | | | **60.38(+0.52†)** | 72.26 | 2.75 | 2% | 1.0 |
| | | | 58.46(+1.98†) | 78.92 | 2.75 | 2% | 1.0 |

[*] **Bolded** entries represent best accuracies/shortest compression time in sparsity class
[†] Accuracy improvements compared to pre-finetuned, SBC pruned models

Table 6: ImageNet pruning on Spike-Driven Transformer: SBC vs. ExactOBS

| Dataset | Arch. | Time step | Top-1 Acc. **SBC (Ours)** (%) | Top-1 Acc. ExactOBS (%) | Weight Sparsity (%) | Time (h) A4500 Eq. | Calib. size (% train) |
|---|---|---|---|---|---|---|---|
| ImageNet | SDT 6_512 | 4 | **73.89** | **73.89** | 0.00 | – | – |
| | | | **71.88** | 71.68 | 50.00 | 0.212 | 1% |
| | | | **64.27** | 63.40 | 66.67 | 0.204 | 1% |
| | | | **50.91** | 49.31 | 75.00 | 0.206 | 1% |

[*] **Bolded** entries represent best accuracies in each sparsity class.

## E  ENERGY USAGE, APPROXIMATED BY SOPS

We use synaptic operations (SOPs) (Shi et al., 2024) as a proxy metric to estimate the energy consumption of an SNN on neuromorphic hardware. This is supported by (Basu et al., 2022), a detailed review of existing neuromorphic hardware. Below, we provide average SOPs for 4Conv-2FC trained on CIFAR10-DVS, VGGSNN-16 trained on CIFAR100, and SEW-ResNet18 trained on ImageNet at different sparsities.

| Dataset | Model | Sparsity (%) | Synaptic ops per forward (M) |
|---|---|---|---|
| cifar10dvs | 4conv2fc | 0 | 119.2 |
| | | 80 | 30.60 |
| | | 90 | 25.51 |
| | | 95 | 20.40 |
| | | 97 | 16.96 |
| | | 98 | 14.53 |
| cifar100 | vggsnn16 | 0 | 107.95 |
| | | 68.30 | 66.31 |
| | | 89.91 | 35.85 |
| | | 95.69 | 20.84 |
| ImageNet | SEW-ResNet-18 | 0 | 1334.4 |
| | | 64.74 | 786.77 |
| | | 70.71 | 716.84 |
| | | 72.26 | 697.24 |
| | | 78.92 | 603.90 |

Table 7: Synaptic operations per forward pass at different sparsity levels.

## F ROBUSTNESS AGAINST SMALL CALIBRATION DATASETS

We evaluated SBC under varying calibration-set sizes on three models: 4Conv2FC on CIFAR-10-DVS, VGGSNN-16 on CIFAR-100, and SEW-ResNet18 on ImageNet. For each setting, we drew three independent stratified calibration sets of the indicated size, applied SBC using only these samples, and reported the mean and standard deviation of the resulting test accuracy. The numbers in the tables denote the absolute calibration-set size.

| Dataset | Model | Calibration size | Accuracy (%) |
|---|---|---|---|
| CIFAR10-DVS | 4CONV2FC | 10 | 49.64 (+-1.27 |
| | | 90 | 61.90 (+-0.21 |
| | | 900 | 63.84 (+-0.49 |
| | | 9000 | 64.04 (+-0.35 |
| CIFAR100 | VGGSNN-16 | 100 | 62.19 (+-0.43 |
| | | 1000 | 65.80 (+-0.07 |
| | | 10000 | 66.42 () |
| ImageNet | SEW-ResNet18 | 100 | 49.35 (+-0.19 |
| | | 1000 | 55.06 (+-0.12 |
| | | 10000 | 55.89 (+-0.11 |
| | | 50000 | 55.89 (+-0.04 |

Table 8: Classification accuracy at different calibration dataset sizes for each dataset/model pair.

## G LLM USAGE

LLM assistance: We used an LLM for copy-editing and formatting only. The authors take full responsibility for the content.

