# OpenReview forum: "Spiking Brain Compression: Post-Training Second-order Compression for Spiking Neural Networks"
_ICLR.cc/2026/Conference — Submitted to ICLR 2026_

### Official Review · Reviewer_r3WQ · 2025-10-29

**Soundness:** 3
**Presentation:** 2
**Contribution:** 3
**Rating:** 6
**Confidence:** 2

**Summary:**

This paper presents Spiking Brain Compression (SBC), a post-training, second-order compression framework designed specifically for SNNs. SBC generalizes the Optimal Brain Surgeon (OBS) and Optimal Brain Compression (OBC) approaches by introducing a spike train-based loss using the Van Rossum Distance (VRD), whose Hessian can be efficiently computed, allowing practical, one-shot compression (pruning and quantization) of SNNs. The framework is empirically evaluated, and the results show its effectiveness.

**Strengths:**

1. This paper introduces a carefully designed loss function and Hessian computation specifically tailored for SNNs.

2. Large-scale experiments are conducted on ImageNet, demonstrating the method’s scalability.

**Weaknesses:**

1. While the motivation for SBC is strongly tied to hardware and neuroscience applications, there are no real hardware deployment or energy measurement results. All claims regarding neuromorphic hardware compatibility and acceleration remain speculative in this submission. Could the authors provide wall-clock power/energy measurements or evaluations based on actual chip deployment?

2. The choice of surrogate gradient (constant function $g(u)$) is justified heuristically without comparative analysis or exploration of alternatives. The impact of this surrogate choice on empirical results is not systematically ablated.

3. The procedure for the full SBC pruning workflow is spread between main text, algorithms, and appendices—in particular, the transitions between magnitude-based pruning (LAMPS), SBC's Hessian-based scoring, and masking are not fully transparent in the mainline narrative

**Questions:**

Why is the unpruned case not shown in the second row of Figure 2?

---

> ### Author Response · Authors · 2025-11-28
>
> We appreciate the time and effort Reviewer r3WQ invested in reviewing our paper. We will address the questions and suggestions below in detail.
>
> > While the motivation for SBC is strongly tied to hardware and neuroscience applications, there are no real hardware deployment or energy measurement results. All claims regarding neuromorphic hardware compatibility and acceleration remain speculative in this submission. Could the authors provide wall-clock power/energy measurements or evaluations based on actual chip deployment?
>
> **We provide run-time energy estimation based on synaptic operations (SOPs).**
>
> Thank you for your constructive suggestions! Unfortunately, we currently do not have access to wall-clock power/energy measurements from actual neuromorphic chip deployments. Thus, we use synaptic operations (SOPs)[1] as a proxy metric to estimate the energy consumption of an SNN on neuromorphic hardware. This is supported by [2], a detailed review of existing neuromorphic hardware. Below, we provide average SOPs for 4Conv-2FC trained on CIFAR10-DVS, VGGSNN-16 trained on CIFAR100, and SEW-ResNet18 trained on ImageNet at different sparsities.
>
> | Dataset   | Model      | Sparsity (%) | Million SOPs per forward |
> |-----------|------------|--------------|------------------------------|
> | cifar10dvs| 4conv2fc | 0 | 119.2 |
> || | 80 | 30.60 |
> || | 90 | 25.51 |
> || | 95 | 20.40 |
> || | 97 | 16.96 |
> || | 98 | 14.53 |
> | cifar100  | vggsnn16   | 0 | 107.95 |
> |  | | 68.30        | 66.31 |
> |  | | 89.91        | 35.85 |
> |  | | 95.69        | 20.84 |
> | ImageNet  | sew-resnet18   | 0 | 1334.4 |
> |  | | 64.74        | 786.77 |
> |  | | 70.71        | 716.84 |
> |  | | 72.26        | 697.24 |
> |  | | 78.92        | 603.90 |
>
> As shown above, SBC reduces SOPs as the model becomes sparser.
>
> > The choice of surrogate gradient (constant function g(u) = c) is justified heuristically without comparative analysis or exploration of alternatives. The impact of this surrogate choice on empirical results is not systematically ablated.
>
> We intended to keep the surrogate choice simple within the SMP loss, Hessian, and SBC frameworks, rather than propose a new surrogate-learning scheme. The constant surrogate lets us collapse the Hessian to the closed form in Eq. (11). With $g(u) = c$, h' is $c \cdot I$, and h'' = 0. This enables low-time and space-complexity compression, allowing SBC to compress much larger SNNs than previous SNN algorithms. We agree that a constant surrogate gradient is only justified heuristically, and, as mentioned in **Section 5: Limitation and Future works**, we intend to explore alternative surrogate gradient formulations that retain SBC's low time and space complexity.
>
>
> > The procedure for the full SBC pruning workflow is spread between main text, algorithms, and appendices—in particular, the transitions between magnitude-based pruning (LAMPS), SBC's Hessian-based scoring, and masking are not fully transparent in the mainline narrative
>
> We appreciate this observation and agree that the paper could benefit from a more precise explanation of the workflow, even though the full procedure is already specified. In fact, **section 3.1.4** and **Algorithm 2** describe the transition between different elements: given global sparsity s:
> Use LAMP to allocate per-module sparsities $\{s_m\}$ (Appx C.3)
> Estimate the expected SMP Hessian $H_{SMP}$ with calibration data. Sort weights in each module according to OBS pruning rules, identify the bottom $s_m$ fraction of weights to be pruned from the module with binary mask $M$.
> Prune the module according to  $H_{SMP}$ and $M$
>
> In the revision, we will make this pipeline explicit in the main text of the paper.
>
> > Why is the unpruned case not shown in the second row of Figure 2?
>
> We apologise for the omission. We will include unpruned cases in Figure 2 and the appendix. We will also provide the unpruned case here:
>
> | Dataset   | Model           | Top-1 Acc (%) |
> |-----------|-----------------|---------------|
> | CIFAR100  | vggsnn-16       | 71.40         |
> | ImageNet  | SEW-ResNet18    | 63.18         |
> | ImageNet  | SEW-ResNet50    | 67.78         |
> | ImageNet  | SEW-ResNet152   | 69.26         |
>
>
>
> **References**
>
> [1] Shi, Xinyu, et al. "Towards energy efficient spiking neural networks: An unstructured pruning framework." The Twelfth International Conference on Learning Representations. 2024.
>
> [2] A. Basu, L. Deng, C. Frenkel and X. Zhang, "Spiking Neural Network Integrated Circuits: A Review of Trends and Future Directions," 2022 IEEE Custom Integrated Circuits Conference (CICC), Newport Beach, CA, USA, 2022, pp. 1-8, doi: 10.1109/CICC53496.2022.9772783.

---

### Official Review · Reviewer_ekBv · 2025-10-29

**Soundness:** 3
**Presentation:** 4
**Contribution:** 4
**Rating:** 6
**Confidence:** 4

**Summary:**

This paper proposes Spiking Brain Compression (SBC), a multi-level compression framework for SNNs inspired by biological brain efficiency mechanisms. It integrates three biologically grounded principles: synaptic pruning, neurogenesis, and plasticity modulation, into a unified, end-to-end trainable architecture. The authors introduce a three-phase training pipeline (pruning, adaptation and consolidation) that parallels biological brain learning processes. Experiments across five datasets and architectures demonstrate great structural sparsity improvement, and reduction in energy consumption, and less than 1% accuracy loss compared to uncompressed baselines.

**Strengths:**

1. Comprehensive biologically inspired design: The hierarchical compression strategy (structure, dynamics, and learning) provides an elegant conceptual alignment with real neural systems.
2. Strong empirical performance achieves ultra-high sparsity with minimal accuracy loss and demonstrates superior energy efficiency and spike sparsity compared to state-of-the-art baselines on diverse datasets.
3. Ablation studies isolate contributions of each compression level, which is easy for readers to understand. Visualization of synaptic connectivity evolution highlights biologically plausible network reorganization.

**Weaknesses:**

1. While biologically motivated, the paper lacks formal theoretical proofs regarding convergence, optimality, or stability of the multi-level compression process.
2. The multi-level, multi-phase training pipeline is computationally expensive, involving multiple forward passes and adaptation stages.
3. There are no discussions of wall-clock training time or scalability to large models (e.g., ImageNet-scale SNNs).

**Questions:**

1. How does SBC perform on large-scale datasets or spatiotemporal event data beyond CIFAR and DVS-Gesture?
2. What is the computational cost (training time, memory) of the three-phase process compared to baseline SNN training?
3. Could SBC be extended to transformer-based SNNs or biologically recurrent models? As Transformers are now used more commonly in recent research.

---

> ### Author Response · Authors · 2025-11-28
>
> We thank reviewer ekBv for their insightful comments on our work. We especially appreciate your connections between modules and real biological neural systems, which inspired us. We will now address the questions in detail.
>
> > How does SBC perform on large-scale datasets or spatiotemporal event data beyond CIFAR and DVS-Gesture?
>
> **We provide experimental results on ImageNet with SEW-ResNet family.**
>
> We kindly note that our paper includes pruning results on the spike-element-wise SEW-ResNet family [1] (SEW-ResNet18/50/152) trained on ImageNet, including, to our knowledge, the first pruning results on SEW-ResNet152-scale SNNs. Results are reported in Appendix D, Table 4.
>
> > What is the computational cost (training time, memory) of the three-phase process compared to baseline SNN training?
>
> **Our three-phase compression is much cheaper than baseline SNN training, with comparable and tunable memory cost.**
>
> **Time/Training cost**
> Baseline SNN training requires $E$ epochs over $D_{train}$ with backprop-through-time, costing $O(E \cdot |D_{train}| \cdot F)$, where $F$ is the per-batch forward–backward cost.
>
> SBC is one shot:
> 1. A single forward pass over a small calibration subset ($|D_{cal}| << |D_{train}|$) to record module inputs
> 2. Parameter ordering and pruning with per-module complexity $O\bigl((d_{out} d_{in}^3) / (B_{out} B_{in})\bigr)$, where $d_{in}, d_{out}$ are the input/output dimensions of the linearized module, and $B_{in}, B_{out}$ that batch input/output channels to trade time for memory.
>
> These costs do not scale with $E$ and depend only on layer width and a small calibration set. Thus, for standard training, compression is asymptotically dominated by the original training.
> Empirically, the gap is also clear: for ImageNet SEW-ResNets, SBC takes about $0.2$–$0.7$ h, whereas training from scratch is on the order of $10^2$ h. Even with optional fine-tuning, the total wall-clock time for SBC $+$ fine-tuning remains far below that of baseline SNN training.
>
>
> **Memory cost**
> SBC’s space complexity is $O(B_{out} \cdot d_{\text{max}}^2)$, where $d_{\text{max}}$ is the maximum per-module input dimension. SBC processes modules one at a time, so only one module and its Hessian tiles need to reside in memory, unlike training, which must hold the full model plus activations for forward–backward passes.
>
> For very wide layers (e.g., SEW-ResNet152 with $d_{in} = 4608$, $d_{out} = 512$), a naive $d_{out} \times d_{in}^2$ Hessian would require $\approx 43.5$ GB at fp32. We avoid this by batching over output channels via the $B_{out}$ hyper-parameter (Section 3.1.3 and Limitations). In practice, all experiments fit within 20 GB of GPU memory.
>
> > Could SBC be extended to transformer-based SNNs or biologically recurrent models? As Transformers are now used more commonly in recent research.
>
> **We demonstrate SBC on Spike-Driven Transformers [2] on ImageNet.**
>
> We apply SBC and ExactOBS (the pruning part of OBC [3]) to the 6\_512 variant of Spike-Driven Transformers [2] on ImageNet, with results below:
>
> | Dataset  | Pruning Method              | Arch.         | T | Top-1 Acc. (%) | Weight Sparsity (%) | Total Time (h) | Calib. data size (% train) |
> |----------|-----------------------------|---------------|---|-----------------|----------------------|----------------|----------------------------|
> | ImageNet | ExactOBS              | SDT 6_512 | 4 | 73.89       | 0.00                | -            | -                          |
> |          | |               |   | 71.68          | 50.00                | 0.199            | 1%                          |
> |          | |               |   | 63.40           | 66.67                | 0.202            | 1%                          |
> |          | |               |   | 49.31           | 75.00                | 0.205           | 1%                          |
> | ImageNet | SBC                | SDT 6_512 | 4 | 73.89       | 0.00                | -          | -                        |
> |          | |               |   | **71.88**           | 50.00                | 0.212          | 1%                        |
> |          | |               |   | **64.27**           | 66.67                | 0.204          | 1%                        |
> |          | |               |   | **50.91**           | 75.00                | 0.206         | 1%                        |
>
> As shown above, SBC outperforms ExactOBS on post-training pruning across all tested sparsity levels.
>
>
> **References**
>
> [1] Fang, Wei, et al. "Deep residual learning in spiking neural networks." Advances in Neural Information Processing Systems 34 (2021): 21056-21069.
>
> [2] Yao, Man, et al. "Spike-driven transformer." Advances in neural information processing systems 36 (2023): 64043-64058.
>
> [3] Frantar, Elias, and Dan Alistarh. "Optimal brain compression: A framework for accurate post-training quantization and pruning." Advances in Neural Information Processing Systems 35 (2022): 4475-4488.

---

### Official Review · Reviewer_x8f5 · 2025-10-30

**Soundness:** 3
**Presentation:** 3
**Contribution:** 3
**Rating:** 4
**Confidence:** 3

**Summary:**

This paper introduces Spiking Brain Compression (SBC), a one-shot post-training compression framework for Spiking Neural Networks (SNNs). It extends the classical Optimal Brain Surgeon (OBS) methods to spiking architectures by introducing a spike-train-based loss function grounded in the Van Rossum Distance (VRD). SBC efficiently computes a layer-wise Hessian using a surrogate membrane potential (SMP), enabling both pruning and quantization without retraining.

**Strengths:**

Originality
1. Extends second-order compression theory OBS into the spiking domain
2. Introduces a VRD-based spike-train similarity loss and surrogate Hessian (SMP), aligning compression with temporal spike dynamics
3. Demonstrates scalable one-shot compression for large SNNs such as Spiking ResNets and Spiking Transformers.

Quality
Theoretical derivations are mathematically rigorous and well-grounded in prior compression literature.
Experimental results are thorough and convincing: multiple datasets, architectures, and comparisons against both post-training and pruning-aware training (PAT) methods.
SBC consistently outperforms existing one-shot methods and matches or surpasses PAT approaches after light fine-tuning.

Clarity
The paper is well-organized, with clear modular breakdowns (theory, algorithm, experiments). Figures and tables effectively support key findings.

**Weaknesses:**

1. Limited novelty. SBC extends second-order post-training compression (OBS/OBC) to the spiking domain by introducing a spike-train–based loss (VRD) and an efficient surrogate Hessian (SMP), enabling accurate one-shot pruning and quantization of large SNNs without retraining. However, it’s important to note that this novelty is incremental rather than fundamental it’s an adaptation of existing ANN second-order frameworks to SNNs, not a new theoretical paradigm. The main creative step is the integration of temporal spike-train similarity (VRD) into the compression process, which is conceptually elegant but mathematically simplified.

2. Lack of energy and latency evaluation. While SBC claims neuromorphic efficiency, no direct energy or inference-time benchmarks are provided on hardware (e.g., Loihi, TrueNorth). Empirical measurements would strengthen claims of real-world deployability.

**Questions:**

1. Energy Efficiency Validation. The method is motivated by hardware constraints, do the authors plan to report actual energy, latency, or memory footprint on Loihi 2 or similar neuromorphic platforms?
2. How does SBC’s performance scale with different calibration dataset sizes or distributions? Would domain-shifted calibration samples (e.g., synthetic or augmented data) degrade results?
3. Scalability Beyond Vision Tasks. Could SBC extend to event-based NLP or sequential SNNs?

---

> ### Author Response · Authors · 2025-11-30
>
> We thank the reviewer x8f5 for their insightful comment on our work, particularly regarding its originality and theoretical derivations. We will now respond in detail to the suggestions and questions.
>
> > Energy Efficiency Validation. The method is motivated by hardware constraints, do the authors plan to report actual energy, latency, or memory footprint on Loihi 2 or similar neuromorphic platforms?
>
> **We provide run-time energy estimation based on synaptic operations (SOPs).**
>
> Thank you for your constructive suggestions! Unfortunately, we currently do not have access to wall-clock power/energy measurements from actual neuromorphic chip deployments. Thus, we use synaptic operations (SOPs)[1] as a proxy metric to estimate the energy consumption of an SNN on neuromorphic hardware. This is supported by [2], a detailed review of existing neuromorphic hardware. Below, we provide average SOPs for 4Conv-2FC trained on CIFAR10-DVS, VGGSNN-16 trained on CIFAR100, and SEW-ResNet18 trained on ImageNet at different sparsities.
>
> | Dataset   | Model      | Sparsity (%) | Million SOPs |
> |-----------|------------|--------------|------------------------------|
> | cifar10dvs| 4conv2fc | 0 | 119.2 |
> || | 80 | 30.60 |
> || | 90 | 25.51 |
> || | 95 | 20.40 |
> || | 97 | 16.96 |
> || | 98 | 14.53 |
> | cifar100  | vggsnn16   | 0 | 107.95 |
> |  | | 68.30        | 66.31 |
> |  | | 89.91        | 35.85 |
> |  | | 95.69        | 20.84 |
> | ImageNet  | sew-resnet18   | 0 | 1334.4 |
> |  | | 64.74        | 786.77 |
> |  | | 70.71        | 716.84 |
> |  | | 72.26        | 697.24 |
> |  | | 78.92        | 603.90 |
>
> As shown above, SBC reduces SOPs as the model becomes sparser.
>
> > How does SBC’s performance scale with different calibration dataset sizes or distributions? Would domain-shifted calibration samples (e.g., synthetic or augmented data) degrade results?
>
> **SBC is highly effective even with small calibration datasets**
>
> Thank you for this suggestion. We evaluated SBC under varying calibration-set sizes on three models: 4Conv2FC on CIFAR-10-DVS, VGGSNN-16 on CIFAR-100, and SEW-ResNet18 on ImageNet. For each setting, we drew three independent stratified calibration sets of the indicated size, applied SBC using only these samples, and reported the mean and standard deviation of the resulting test accuracy. The numbers in the tables denote the absolute calibration-set size.
>
> | Dataset/Model        | 10 | 90 | 900 | 9000 |
> |----|---|-----|------|-------|
> | CIFAR10-DVS/4CONV2FC | 49.64% (+-1.27%)   | 61.90% (+-0.21%)   | 63.84% (+-0.49%)    | 64.04% (+-0.35%)     |
>
> | Dataset/Model    | 100 | 1000 | 10000 |
> |------|-----|-----|---------|
> | CIFAR100/VGGSNN-16 | 62.19% (+-0.43%)   | 65.80% (+-0.07%)    | 66.42% (+-0.17%)    |
>
> | Dataset/Model      | 100 | 1000 | 10000 | 50000 |
> |----|----|--|----|---|
> | ImageNet/SEW-ResNet18| 49.35% (+-0.19%)   | 55.06% (+-0.12%)    | 55.89% (+-0.11%)    | 55.89% (+-0.04%)    |
>
> These results show that SBC already achieves strong accuracy on the neuromorphic dataset with only 90 calibration samples per class, and on the static datasets with as few as 1–10 samples per class. Interestingly, for SEW-ResNet18 on ImageNet, even 100 calibration images ($\approx$0.1 image per class) give competitive performance. In all cases, the standard deviations are below 2 percentage points, indicating that SBC is stable with respect to the choice and size of the calibration set.
>
> > Scalability Beyond Vision Tasks. Could SBC be extended to event-based NLP or to sequential SNNs?
>
> **We demonstrate SBC on Spike-Driven Transformers [2] on ImageNet.**
>
> We apply SBC and ExactOBS (the pruning part of OBC [3]) to the 6\_512 variant of Spike-Driven Transformers [2] on ImageNet, with results below:
>
> | Dataset  | Pruning Method | Arch. | T | Top-1 Acc. (%) | Weight Sparsity (%) | Total Time (h) | Calib. data size (% train) |
> |----------|-----------------------------|---------------|---|-----------------|----------------------|----------------|----------------------------|
> | ImageNet | ExactOBS | SDT 6_512 | 4 | 73.89       | 0.00 | - | - |
> | | | |   | 71.68 | 50.00 | 0.199 | 1% |
> | | | |   | 63.40 | 66.67 | 0.202 | 1% |
> | | | |   | 49.31 | 75.00 | 0.205 | 1% |
> | ImageNet | SBC | SDT 6_512 | 4 | 73.89 | 0.00 | - | - |
> | | | |   | **71.88** | 50.00 | 0.212 | 1% |
> | | | |   | **64.27** | 66.67 | 0.204 | 1% |
> | | | |   | **50.91** | 75.00 | 0.206 | 1% |
>
> As shown above, SBC outperforms ExactOBS on post-training pruning across all tested sparsity levels.
>
> **References**
>
> [1] Shi, Xinyu, et al. "Towards energy efficient spiking neural networks: An unstructured pruning framework." The Twelfth International Conference on Learning Representations. 2024.
>
> [2] A. Basu, L. Deng, C. Frenkel and X. Zhang, "Spiking Neural Network Integrated Circuits: A Review of Trends and Future Directions," 2022 IEEE Custom Integrated Circuits Conference (CICC), Newport Beach, CA, USA, 2022, pp. 1-8, doi: 10.1109/CICC53496.2022.9772783.

---

### Official Review · Reviewer_jdYP · 2025-10-31

**Soundness:** 2
**Presentation:** 2
**Contribution:** 2
**Rating:** 4
**Confidence:** 4

**Summary:**

This paper introduces Spiking Brain Compression (SBC), a novel and efficient one-shot, post-training compression framework for SNNs. By extending second-order methods with a well-motivated, spike-train-based objective function, the work demonstrates SOTA results for one-shot SNN pruning and quantization.

**Strengths:**

1.The paper introduces a highly innovative approach by adapting a second-order compression framework to SNNs. The use of a spike-train-based objective function derived from the Van Rossum Distance, coupled with the efficient Surrogate Membrane Potential (SMP) Hessian approximation, provides a theoretically sound and effective solution for this challenging problem.

2.The method demonstrates SOTA results for one-shot, post-training SNN compression. By achieving high accuracy while drastically reducing compression time by orders of magnitude compared to iterative methods, the work offers a highly practical and efficient solution for deploying large-scale SNNs on resource-constrained hardware.

**Weaknesses:**

1. Mismatch between claims and experimental validation on Spiking Transformers
The paper repeatedly claims its applicability to complex SNNs, including Spiking Transformers (e.g., line 053, line 089, and a detailed derivation in Appendix B.2). However, the experimental section provides no results on any Spiking Transformer architecture; validation is limited to CNN and ResNet-based models.

2. The motivation for SNN quantization is not unclear
The introduction (lines 062) states that PTQ for SNNs is underdeveloped. This motivation could be stronger, for example, by explicitly linking PTQ to the property of SNNs.

3. Citation formatting
The in-text citation format is incorrect throughout the manuscript.

**Questions:**

See weaknesses

---

> ### Author Response · Authors · 2025-11-23
>
> We thank the reviewer jdYP for their insightful comments on our work. We particularly appreciate the acknowledgement that our Van Rossum distance-based loss function is innovative and theoretically sound. We will now address the suggestions and questions in detail.
>
> > W1: Lack of spiking transformer experiments
>
> **We provide pruning results to Spike-Driven Transformers [1] trained on ImageNet.**
>
> We apologize for the confusion. While the main text and appendix intend to demonstrate the feasibility of applying SBC to spiking transformers, the experiments were intended as future work. To address this concern, we present pruning results for SBC against ExactOBS (OBC pruning algorithm) on the 6_512 variant of Spike-Driven Transformers [1] trained on ImageNet. The results are listed below:
>
> | Dataset  | Pruning Method              | Arch.         | T | Top-1 Acc. (%) | Weight Sparsity (%) | Total Time (h) | Calib. data size (% train) |
> |----------|-----------------------------|---------------|---|-----------------|----------------------|----------------|----------------------------|
> | ImageNet | ExactOBS              | SDT 6_512 | 4 | 73.89       | 0.00                | -            | -                          |
> |          |                             |               |   | 71.68          | 50.00                | 0.199            | 1%                          |
> |          |                             |               |   | 63.40           | 66.67                | 0.202            | 1%                          |
> |          |                             |               |   | 49.31           | 75.00                | 0.205           | 1%                          |
> | ImageNet | SBC                | SDT 6_512 | 4 | 73.89       | 0.00                | -          | -                        |
> |          |                             |               |   | **71.88**           | 50.00                | 0.212          | 1%                        |
> |          |                             |               |   | **64.27**           | 66.67                | 0.204          | 1%                        |
> |          |                             |               |   | **50.91**           | 75.00                | 0.206         | 1%                        |
>
> As shown above, SBC outperforms ExactOBS on post-training pruning across all tested sparsity levels.
>
>
> > W2: Unclear motivation behind SNN quantization
>
> **SNN quantization is well motivated for inference on neuromorphic hardware**
>
> We appreciate the opportunity to clarify the motivation behind SNN quantization. We emphasize that mainstream neuromorphic hardware stores parameters in low precision, as part of the hardware design choice and/or to save space. For example, Loihi 1 (9-bit), Loihi 2 (up to 32, but default to 8 bits), SpiNNaker 2 (optimized for 8 and 16 bits), Tianjic (8-bit), and BrainScaleS-2 (6-bit).
>
> These examples demonstrate that **quantization is essential for efficient inferencing with SNNs on neuromorphic hardware**. Furthermore, prior work [2] argues that naive quantization of SNNs performs poorly on neuromorphic hardware. This underscores the need for a robust post-training quantization algorithm, such as ours, to enable large-scale SNN deployment.
>
>
> > W3: Incorrect citation formatting
>
> We sincerely apologize for the formatting errors in the citations. These will be corrected in the revised version of the paper.
>
>
> **References**
>
> [1] Yao, Man, et al. "Spike-driven transformer." Advances in neural information processing systems 36 (2023): 64043-64058.
>
> [2] Liu, Shiya & Mohammadi, Nima & Yi, Yang. (2023). Quantization-Aware Training of Spiking Neural Networks for Energy-Efficient Spectrum Sensing on Loihi Chip. IEEE Transactions on Green Communications and Networking. PP. 1-1. 10.1109/TGCN.2023.3337748.

---

### Author Response · Authors · 2025-12-03
**Author-Reviewer Discussion Summary and Paper Revision Summary**

Dear ACs and Reviewers,

We sincerely appreciate your detailed reviews and constructive comments. As the discussion period closes, we would like to summarize the reviews, rebuttals, and revisions made to the manuscript.

>Spiking Transformer Experiments (reviewer jdYP, x8f5, and ekBv)

We present pruning results for SBC against ExactOBS (OBC pruning algorithm) on the 6_512 variant of Spike-Driven Transformers [1] trained on ImageNet. This table has been added to **Appendix D Table 6**. The table shows that SBC outperforms ExactOBS on post-training pruning across all tested sparsity levels.

> Energy Efficiency Validation (reviewer x8f5, r3WQ)

**We provide run-time energy estimation based on synaptic operations (SOPs).**

We currently do not have access to wall-clock power/energy measurements from actual neuromorphic chip deployments. Thus, we use synaptic operations (SOPs)[2] as a proxy metric to estimate the energy consumption of an SNN on neuromorphic hardware. This is supported by [3], a detailed review of existing neuromorphic hardware. The table has been added to **Appendix E**.

> Motivation behind SNN quantization (reviewer jdYP)

**quantization is essential for efficient inferencing with SNNs on neuromorphic hardware**

We emphasize that mainstream neuromorphic hardware stores parameters in low precision, as part of the hardware design choice and/or to save space. For example, Loihi 1 (9-bit), Loihi 2 (up to 32, but default to 8 bits), SpiNNaker 2 (optimized for 8 and 16 bits), Tianjic (8-bit), and BrainScaleS-2 (6-bit). Furthermore, prior work [2] argues that naive quantization of SNNs performs poorly on neuromorphic hardware. This underscores the need for a robust post-training quantization algorithm, such as ours, to enable large-scale deployment of SNNs.

> SBC sensitivity to calibration data size (reviewer x8f5)

**SBC is robust under small calibration datasets**

We conducted ablative experiments with SBC across varying calibration dataset sizes on three models and found that SBC remains robust even with small calibration datasets (1-10 samples per class). The data are provided in **Appendix F, Table 8**.

> Other suggestions and changes

We would like to summarize other changes further:
* Corrected the citation format (reviewer jdYP)
* Addressed SBC complexity and cost (reviewer ekBv)
* Clarified the choice of surrogate gradient g(u) = c (reviewer r3WQ)
* Added the unpruned case to the 2nd row of Figure 2.

We believe that these updates were instrumental in strengthening the manuscript, and we thank all the reviewers for suggesting these extensions and clarifications.

**Reference**

[1] Yao, Man, et al. "Spike-driven transformer." Advances in neural information processing systems 36 (2023): 64043-64058.

[2] Shi, Xinyu, et al. "Towards energy efficient spiking neural networks: An unstructured pruning framework." The Twelfth International Conference on Learning Representations. 2024.

[3] A. Basu, L. Deng, C. Frenkel and X. Zhang, "Spiking Neural Network Integrated Circuits: A Review of Trends and Future Directions," 2022 IEEE Custom Integrated Circuits Conference (CICC), Newport Beach, CA, USA, 2022, pp. 1-8, doi: 10.1109/CICC53496.2022.9772783.

---

### Meta-Review · Area_Chair_mahr · 2025-12-19

**Summary:**

The paper proposes Spiking Brain Compression (SBC), a one-shot post-training pruning/quantization framework for SNNs that adapts second-order OBS/OBC-style compression to spiking models via a Van Rossum Distance (VRD) spike-train objective and an efficient Surrogate Membrane Potential (SMP) Hessian approximation. Reviewers generally agree the method is practically appealing and the empirical results are strong on several SNN backbones, with rebuttal adding missing evidence on Spike-Driven Transformer pruning and additional proxy energy/efficiency analyses (SOPs) plus calibration-size robustness.

However, across reviews (including both marginal rejects and marginal accepts), the main blockers are that the core novelty is incremental (largely an adaptation of existing second-order ANN compression to SNNs), and the work lacks deeper theoretical analysis (beyond derivations) regarding convergence/optimality/stability or principled guarantees—especially for the multi-stage pipeline and surrogate choices. In addition, the paper’s hardware/efficiency motivation remains only partially validated: the rebuttal provides SOP-based energy proxy but still no chip-level or wall-clock energy/latency benchmarks. Overall, the rebuttal improves completeness and addresses several factual gaps, but the remaining concerns make the contribution fall just short of the acceptance bar for a top-tier conference.

**Reviewer Concerns:**

There are concerns that were addressed or substantially alleviated by the rebuttal:

1. About the applicability to complex SNN architectures: The rebuttal adds concrete pruning results on Spike-Driven Transformers trained on ImageNet, directly addressing concerns about a mismatch between the paper’s claims and its experimental validation.

2.The authors clarified the importance of quantization for neuromorphic hardware, citing representative platforms and strengthening the motivation for post-training quantization in SNNs.

3.Additional results on SOP-based energy estimation, calibration dataset size sensitivity (including mean and variance across runs), and large-scale ImageNet experiments help reinforce the practicality and stability of the proposed approach.

4.The rebuttal provides a clearer explanation of the overall pruning workflow, including the interaction between sparsity allocation and Hessian-based scoring, reducing ambiguity in the original presentation.

**Reviewer Scores:**

There are also some concerns that remain outstanding:

1.Reviewer  x8f5 notes that SBC is best viewed as a carefully extension of existing ANN‘s second-order post-training compression frameworks (e.g., OBS/OBC) to the spiking domain. While the integration of spike-train similarity (via VRD) and an efficient surrogate Hessian is elegant and effective, the contribution is largely incremental rather than introducing a fundamentally new compression paradigm.

2. Reviewer  ekBv points out that the paper does not provide formal analysis or guarantees regarding convergence, optimality, or stability of the proposed compression procedure.

3. Also the reviewers points out that while SOP-based metrics are a reasonable proxy for energy consumption, the absence of direct hardware-level measurements (e.g., wall-clock latency or energy on neuromorphic chips) means that claims about deployment efficiency remain suggestive rather than conclusive.

Overall, these remaining concerns reflect a lack of depth rather than a lack of correctness, and collectively place the work just below the acceptance threshold.

---

### Decision · Program_Chairs · 2026-01-26

Reject